# Procedures in Fecal Microbiota Transplantation for Treating Irritable Bowel Syndrome: Systematic Review and Meta-Analysis

**DOI:** 10.3390/jcm12051725

**Published:** 2023-02-21

**Authors:** Tânia Rodrigues, Sofia Rodrigues Fialho, João Ricardo Araújo, Rita Rocha, André Moreira-Rosário

**Affiliations:** 1NOVA Medical School, Faculdade de Ciências Médicas, NMS, FCM, Universidade NOVA de Lisboa, 1169-056 Lisboa, Portugal; 2CINTESIS@RISE, NOVA Medical School, Faculdade de Ciências Médicas, NMS, FCM, Universidade NOVA de Lisboa, 1169-056 Lisboa, Portugal; 3i3S—Instituto de Investigação e Inovação em Saúde, Universidade do Porto, 4200-135 Porto, Portugal

**Keywords:** irritable bowel syndrome, fecal microbiota transplantation, microbiome therapy, meta-analysis, randomized controlled trials

## Abstract

Background: Irritable bowel syndrome (IBS) is a prevalent gastrointestinal disease with no effective treatment. Altered microbiota composition seems implicated in disease etiology and therefore fecal microbial transplantation (FMT) has emerged as a possible treatment therapy. To clarify the clinical parameters impacting FMT efficacy, we conducted a systematic review with subgroup analysis. Methods: A literature search was performed identifying randomized controlled trials (RCTs) comparing FMT with placebo in IBS adult patients (8-week follow-up) with a reported improvement in global IBS symptoms. Results: Seven RCTs (489 participants) met the eligibility requirements. Although FMT seems not to be effective in global improvement of IBS symptoms, subgroup analysis shows that FMT through gastroscopy or nasojejunal tube are effective IBS treatments (RR 3.03; 95% CI 1.94–4.73; I^2^ = 10%, *p* < 0.00001). When considering non-oral ingestion routes, IBS patients with constipation symptoms are more likely to benefit from FMT administration (*p* = 0.003 for the difference between IBS subtypes regarding constipation). Fresh fecal transplant and bowel preparation seem also to have impact on FMT efficacy (*p* = 0.03 and *p* = 0.01, respectively). Conclusion: Our meta-analysis revealed a set of critical steps that could affect the efficacy of FMT as clinical procedure to treat IBS, nevertheless more RCTs are needed.

## 1. Introduction

Irritable bowel syndrome (IBS) is a symptom-based functional bowel disorder characterized by abdominal pain and altered bowel habits in the absence of detectable structural or biochemical abnormalities [1]. With a prevalence of approximately 4–10% worldwide [2,3], IBS is one of the most prevalent gastrointestinal (GI) disorders and a cause of substantial burden to healthcare services and society [4]. Due to its relapsing and chronic nature, this condition impacts patients’ social interactions and quality of life (QOL) [5]. Despite this, current treatments for IBS are often inadequate and, unexpectedly, the pipeline for developing new treatments is relatively poor [6].

According to the gold standard symptom-based diagnostic criteria for IBS, the Rome criteria [1], IBS is classified into 4 subtypes: diarrhea-predominant type (IBS-D), constipation-predominant type (IBS-C), mixed type (IBS-M) or unclassified type (IBS-U) (Appendix A). Since no specific biomarkers are available to distinguish between different IBS subtypes, criteria is based on the abdominal pain and stool form changes, as assessed by the Bristol Stool Form Scale [7] (Appendix A).

A multifactorial etiology has been associated to IBS, involving a complex interaction between genetic, psychological and environmental factors that lead to GI motility dysfunction and altered visceral sensations [8]. Some studies have shown an altered microbiota composition in patients with IBS, supporting an important role for the intestinal microbiota in IBS etiology [9,10,11,12]. Since intestinal microbiota play a key role in intestinal immunity and inflammation [13,14], manipulation of its composition has been proposed as a treatment strategy for IBS.

Fecal microbial transplantation (FMT) is a technique in which fecal material containing gut microorganisms are transferred from a healthy donor to a patient, with the intention of correcting imbalances in the microbial community of the gut. FMT can be administered either directly to the colon—via colonoscopy, or less frequently via flexible sigmoidoscopy or an enema—or to the upper gastrointestinal tract via nasoenteric tubes, gastroscopy, or capsule ingestion [15].

Based on the concept of repopulating intestinal microbiota, FMT has been proven effective for the treatment of recurring *Clostridioides difficile* infection (CDI), by inhibiting its colonization and, so far no major differences have been found between the different FMT delivery modes [15]. However, it remains unclear whether FMT efficacy extends to other gastrointestinal disorders such as IBS. To our best knowledge, eight systematic reviews with meta-analysis have been conducted for evaluating the efficacy of FMT on IBS treatment up-to-now [16,17,18,19,20,21,22,23], four of them in 2022. This number shows the clinical relevance of IBS as well as FMT as therapy. Previous systematic reviews have been consistent in unveiling the route of FMT administration as the major factor that impacts FMT efficacy, however they have not explored in detail other clinical and technical conditions influencing FMT on IBS treatment. This exploitation will be critical for designing novel randomized clinical trials addressing FMT as an intervention procedure for treating patients with IBS. Knowing this, we decided to conduct a systematic review with subgroup analysis for assessing the methodological conditions that are more likely to impact FMT efficacy. This knowledge may allow the optimization of the FMT procedure with potential positive impact on its clinical efficacy for IBS treatment.

## 2. Materials and Methods

### 2.1. Protocol and Registration

This study was developed in accordance with the preferred reporting items for systematic reviews and meta-analyses (PRISMA) statement [24] and the Cochrane Handbook for Systematic Reviews of Interventions [25] guidelines. The protocol was registered in the International Prospective Register of Systematic Reviews (PROSPERO) with the registration code CRD42021252141.

### 2.2. Selection Criteria

We defined inclusion and exclusion criteria in accordance to the PICO (Population, Intervention, Comparator and Outcomes) [26] strategy. Inclusion criteria were: (1) prospective, randomized, double-blind, placebo-controlled trials (parallel group or first arm of cross-over); (2) with adult patients older than 16 years with IBS defined by accepted symptom-based criteria including Manning, Kruis, Rome I, Rome II, Rome III, or Rome IV (Population); (3) compared FMT (Intervention) with placebo consisting of only the FMT excipients or an autologous FMT (Comparator); (4) reported improvement in global IBS symptoms (Outcome); and (5) with a minimum duration of 8-week follow-up, according to the recommended duration for the assessment of short-term response to therapy in functional GI disorders [27].

Review articles, systematic reviews, meta-analysis, letters, conference abstracts, case reports, case series, position papers, and author’s replies were excluded. Only studies published in English were included.

### 2.3. Search Strategy

To identify eligible reviews, we searched on Cochrane, MEDLINE, Scopus and Web of Science databases on 27 July 2021. Both medical subject headings (MeSH) terms and free text terms referring to fecal microbiota transplantation combined with terms referring to irritable bowel syndrome were used. The PubMed search strategy was converted to search in other databases (Appendix A).

### 2.4. Study Selection

We used the online tool Rayyan [28] to remove duplicates and to screen articles for eligibility, according to the screening criteria. Two independent reviewers (S.F. and T.R.) screened the titles and abstracts of the articles for relevance, and full-text articles were reviewed when title and abstract did not provide enough information. Once potentially relevant studies were identified, full-text articles were then assessed for eligibility according to previously established criteria. Excluded trials and the reasons for exclusion were recorded and any disagreement between reviewers was resolved through discussion. 

### 2.5. Data Extraction

Data items were extracted by two authors for each study; first author, year of publication, country of origin, sample characteristics, methods, and outcomes. Data regarding the global improvement in IBS symptoms, was extracted as intention-to-treat analyses (with dropouts assumed to be non-responders to FMT) and synthesized into tables. When information was missing or incomplete, the corresponding authors were contacted requesting further information. 

### 2.6. Risk of Bias Assessment in Individual Studies

Risk of bias in individual studies was assessed using the updated Cochrane Risk of Bias (RoB 2.0) tool recommended by Cochrane Collaboration [29]. The following five domains were assessed: (1) bias due to the randomization process, (2) bias due to deviations from intended interventions, (3) bias due to missing outcome data, (4) bias in the measurement of the outcome, and (5) bias in the selection of the reported result. Regarding the evaluation of the third domain (missing bias), 10% missing and missing above 5% with imbalances between arms, were classified with “some concerns”. The overall risk of bias was classified as; high risk, having some concerns, and low risk. Reviewers were blinded to each other’s assessment, and disagreements were solved by reaching a consensus.

### 2.7. Quantitative Synthesis

Relative risk (RR) was used as an effect measure for the dichotomous variable “treatment responders”. Effect measures were reported along with the 95% confidence interval (CI). The heterogeneity was assessed through the Cochran’s Q (significance level of 0.1) and I^2^ tests, and when detected, subgroup analysis was performed to explore possible causes. According to the Cochrane guidelines [25], the I^2^ values were interpreted as follows: 0% to 40% might not be important; 30% to 60% may represent moderate heterogeneity; 50% to 90% may represent substantial heterogeneity; 75% to 100% represent considerable heterogeneity.

Pooled estimates were computed and weighted using generic inverse-variance with random-effect. A *p*-value < 0.05 was considered as statistically significant. Statistical analysis was performed using Review Manager (RevMan), version 5.4, The Cochrane Collaboration, 2020.

### 2.8. Grading the Evidence

Funnel plots were used to assess evidence of publication bias. Quality assessment of the evidence for each outcome was scored using Grading of Recommendations Assessment, Development and Evaluation (GRADE) [30]. The meta-analysis was scored with a maximum of 10 points, according to (1) risk of bias, (2) precision, (3) heterogeneity, (4) directness, (5) publication bias, (6) funding bias, (7) effect-size, and (8) dose–response. Based on the final score, we classified the quality of the evidence as high, moderate, low, or very low.

## 3. Results

### 3.1. Study Selection

The literature search identified 5866 citations, of which 267 were reviewed based on eligibility criteria; 243 of the reviewed references were excluded. Of the 24 remaining citations, 17 were excluded after meticulous full-text review, as detailed in Figure 1. In the end, 7 RCTs [31,32,33,34,35,36,37] (full manuscripts) were eligible and included in our meta-analysis.

### 3.2. Study Characteristics

Detailed characteristics of the included RCTs are summarized in Table 1 and Table 2. The sample size of each study ranged from 17 to 165 participants, totalling 489 adults. However, since none of the studies reported a true intention-to-treat analysis, only 465 were analyzed with a total of 298 patients allocated (298 to intervention and 140 to control). 

The diagnosis criteria were Rome III or Rome IV. One study included IBS-D only [31], two studies included IBS-D and IBS-M [35,36] and four studies included all 4 subtypes of IBS [32,33,34,37]. FMT was administered using colonoscopy [34,36,37], gastroscopy [32], nasojejunal tube [35], and oral capsules [31,33]. The 5 nonoral ingestion route studies performed single-dose administration of donor or autologous fecal microbiota preparation whilst the two oral capsule FMT studies used multiple doses (3 and 12 doses) of donor fecal microbiota or placebo consisting of FMT excipients alone (no microbiota). The follow-up time varied between 4 months [32], 6 months [31,33,34], and 12 months [35,36,37]. As first outcome, all studies aimed to evaluate the improvement in gastrointestinal symptoms after transplantation, identified by a decrease in IBS Severity Symptom Scale (IBS-SSS) of 75 or more points at 12 weeks [36], a decrease of 50 or more points at 12 weeks [31,32,33,37], and by other tools [34,35].

### 3.3. Risk of Bias Assessment

According to the Cochrane Collaboration tool [29] three RCTs presented some concerns and one was classified as having high risk of bias (Appendix A). According to funnel plot analysis, there is no evidence of publication bias (Appendix A). However, it should be highlighted that considering the few numbers of studies included in this review, there are probably studies that were not published. We found 3 studies registered on Clinicaltrials.gov, completed more than 18 months ago, that have not published their results yet.

### 3.4. GRADE Assessment

Based on the GRADE assessment (Table 3), the current quality of evidence was “very low” mainly due to the serious risk of bias based on the imprecision of effect estimation. The heterogeneity in the methodology of FMT and placebo interventions between studies also affected the quality, especially in studies with capsule administration. 

### 3.5. IBS Symptoms Improvement

From the 489 participants allocated, 465 were included in the analysis of the primary outcome with a symptoms response rate of 66% (185/282) in patients assigned to FMT, and 41% in patients assigned to placebo (75/183), at 12 weeks of follow-up (Appendix A). Considering an intention-to-treat approach, the clinical response rate at 12 weeks was 62% (185/298) in the FMT group, and 39% in the placebo group (75/191). No significant difference in global improvement of IBS symptoms was observed between groups (RR 1.35; 95% confidence interval (CI) 0.75–2.43, *p* = 0.31 from random effects). Moreover, a significant heterogeneity was identified across all studies (I^2^ = 82%) (Figure 2). Given these results, intention-to-treat subgroup analyses were performed to further explore possible heterogeneity sources (Table 3 and Table 4).

#### 3.5.1. Delivery Method

Subgroup analyses found that delivery method significantly influences the efficacy of FMT in IBS treatment (*p* = 0.0003, for subgroup differences) (Table 4). Accordingly, FMT was associated with symptoms improvement compared with placebo (RR 3.03; 95% CI 1.94–4.73, I^2^ = 10%) in gastroscopy and nasojejunal tube [32,35]. By contrast, no significant improvement was found in colonoscopy [34,36,37] and oral capsules [31,33] (RR 1.36; 95% CI 0.93–1.9; I^2^ = 0% and RR 0.61; 95% CI 0.30–1.22, I^2^ = 64%, respectively) (Figure 3).

#### 3.5.2. Dose

To assess a dose-response in FMT efficacy we separated RCTs that used ≥50 g of fecal material from the remaining studies. Subgroup analyses showed that dose does not significantly influence the efficacy of FMT in IBS treatment (*p* = 0.76, for subgroup differences) (Table 4). In fact, FMT using a dose ≥50 g of fecal material [32,33,36] did not show significant improvement in comparison with a dose lower than 50 g [31,32,34,35,37] (RR 1.35; 95% 0.42–4.37; I^2^ = 93% and RR 1.67; 95% 0.89–3.15; I^2^ = 74%, respectively).

Moreover, even if studies with capsules were excluded from the subgroup analysis (Table 5), the dose effect remained non-statistically significant (*p* = 0.8, for subgroup differences) with a high heterogeneity in both groups, ≥50 g and <50 g of fecal material (RR 2.38; 95% CI 0.96–5.87; I^2^ = 86% and RR 2.06; 95% CI 1.10–3.88; I^2^ = 61%, respectively).

#### 3.5.3. Fresh vs. Frozen

Subgroup analyses found that freezing fecal samples does not significantly influence the efficacy of FMT in IBS treatment (*p* = 0.44, for subgroup differences) (Table 4). However, when fresh fecal samples were used [34,35] FMT was associated with symptom improvement compared with placebo (RR 2.28; 95% CI 1.09–4.78; I^2^ = 0%), while when frozen [31,32,33,37] and both frozen and fresh [36] feces were used no significant improvements were found (RR 1.08; 95% CI 0.43–2.72; I^2^ = 90% and RR 1.50; 95% CI 0.92–2.44, respectively).

Subgroup analysis excluding studies with capsules also found that freezing fecal samples does not significantly influence the efficacy of FMT in IBS treatment (*p* = 0.64, for subgroup differences) (Table 5). However, when fresh fecal samples were used [34,35] FMT was associated with global symptom improvement compared with placebo (RR 2.28; 95% CI 1.09–4.78; I^2^ = 0%, *p* = 0.03), while when frozen [31,32,33,37] and both frozen and fresh [36] feces were used no significant improvements were found (RR 1.94; 95% CI 0.59–6.40; I^2^ = 89% and RR 1.50; 95% CI 0.92–2.44, respectively).

#### 3.5.4. Bowel Preparation

Subgroup analyses showed that bowel preparation does not significantly influence the efficacy of FMT in IBS treatment (*p* = 0.8 for the difference between subgroups) (Table 4). In both groups, with bowel preparation [33,34,35,36] and without bowel preparation [31,32,37], FMT was not associated with symptom improvement compared with placebo (RR 1.26; 95% CI 0.53–2.97; I^2^ = 80%, and RR 1.48; 95% CI 0.58–3.75; I^2^ = 88%, respectively).

When studies with capsules were excluded from the subgroup analysis (Table 5), the influence of bowel preparation on FMT efficacy remained non-statistically significant (*p* = 0.84 for the difference between subgroups). However, the results showed that when bowel preparation was made [34,35,36], FMT was associated with a significant symptom improvement compared with placebo (RR 1.70; 95% CI 1.13–2.55; I^2^ = 0%; *p* = 0.01), while without bowel preparation [32,37] no significant improvement was found (RR 1.94; 95% CI 0.59–6.40; I^2^ = 89%).

#### 3.5.5. IBS Subtype

Two RCTs [32,33] performed subgroup analysis based on IBS subtype and found no differences in the response rate at 12 weeks between the IBS subtypes. Since only two studies grouped efficacy data for different IBS subtypes, we divided them in two groups based on [36] the presence or absence of constipation: with constipation type [32,33,34] and without constipation type [31,35,36,37].

Subgroup analyses showed that IBS subtype does not significantly influence the efficacy of FMT in IBS treatment (*p* = 0.77 for the difference between subgroups) (Table 4). In both groups, with and without constipation type, FMT was not associated with symptoms improvement compared with placebo (RR 1.61; 95% CI 0.30–8.69; I^2^ = 93%, and RR 1.25; 95% CI 0.87–1.79; I^2^ = 31%, respectively).

However, when studies with capsules were excluded, subgroup analyses found that IBS subtype significantly influences the efficacy of FMT in IBS treatment (*p* = 0.003 for the difference between subgroups) (Table 5). Indeed, FMT was associated with higher symptom improvement outcomes when administrated to patients with IBS subtypes with constipation (RR 3.50; 95% CI 2.19–5.60; I^2^ = 0%; *p* < 0.001).

### 3.6. Safety of FMT in IBS

Complete adverse events (AEs) data were available for five studies [32,33,34,36,37]. After pooling data from the five studies, 53 (23%) of 231 patients assigned to FMT reported at least one adverse event, compared with 44 (30%) of 147 allocated to placebo. No significant difference in the total number of AEs was observed in patients receiving FMT compared to control patients (RR 0.91; 95% CI 0.58–1.41), with moderate heterogeneity between studies (I^2^ = 47%, *p* = 0.67) (Figure 4).

In total, four participants had serious AEs. Two patients developed diverticulitis 2 months after FMT (both had diverticulosis verified by colonoscopy and experienced several diverticulitis attacks before FMT) [36], one participant had transient vertigo and nausea after the FMT procedure, requiring a few hours of observation in the hospital [36] and one patient died by suicide during the follow-up [35].

## 4. Discussion

### 4.1. Summary of Evidence

We conducted a systematic review and meta-analysis to identify and explore critical steps in the FMT procedure that must be controlled for the efficacy of this therapy in the treatment of IBS patients.

Using the global improvement in IBS symptoms at 12 weeks after FMT as an endpoint, 7 RCTs involving 489 participants were statistically inconclusive mainly due to high heterogeneity. To explore the methodological factors that may have contributed to this heterogeneity, we carried out multiple subgroup analysis targeting the following variables: delivery method, dosage, fresh versus frozen stool, bowel preparation and IBS subtypes.

Regarding the FMT administration method, FMT through multiple-dose oral liquid capsules [31,33] or colonoscopy [34,36,37] showed no benefit, while FMT via gastroscopy and nasojejunal tubes [32,35] demonstrated a clinical benefit in global IBS symptom improvement compared to placebo. The difference observed between FMT capsule and non-oral ingestion could be due to microbial viability disparities in the FMT content after delivery. Indeed, a higher bacterial viability is expected when donor microbiota is directly released in the gastrointestinal tract of the receiver. Nevertheless, other methodological shortcomings may have contributed to the low efficacy of studies with colonoscopy and capsule administration. Namely, considering the FMT administrated by colonoscopy, two of the three studies [34,36] used different cut-offs for treatment response, which may have led to an underestimation in the efficacy of these RCTs. Responses were defined by a decrease of more than 75 points assessed by IBS-SSS in Johnsen et al. [36], and at least 30% in the total GSRS-IBS symptom score in Holster et al. [34]. Considering capsule administration, in Aroniadis et al. [31] less than 30 g was administered despite 30 g being the dose of fecal transplant recommended by the European Committee on Organ Transplantation [38] and the European Consensus [39]. Furthermore, in Halkjaer et al. [33] final fecal suspensions were stored at −20 °C, when the aforementioned guidelines recommend storage at −80 °C to avoid enzyme activity that can lead to degradation of sensitive microbial populations (e.g., Bacteroidetes) [40]. To avoid these disparities, FMT should be produced in a stool banking center following the European and International consensus guidelines for Good Manufacturing Practices (GMP) of FMT donations [38].

Considering the FMT dosage, only three of the seven RCTs analyzed in this study, used a dose of 50 g or more [32,33,36]. Our meta-analysis shows that higher dose (≥50 g) did not result in greater improvement in IBS global symptoms compared with a lower dose per transplant. However, due to the low number of RCTs included in this meta-analysis and the higher heterogeneity yielded (93%), it is difficult to draw definitive conclusions regarding adequate FMT dose. 

When comparing fresh versus frozen FMT, our meta-analysis found a significant improvement in IBS symptoms when patients received fresh donor stool. However, as already mentioned by Wu et al. [18] interpretation of this result should be done cautiously since the fresh FMT was exclusively delivered through colonoscopy and nasojejunal tube, and that there was high heterogeneity (I^2^ = 90%) among studies using frozen FMT. Thus, the efficacy of frozen FMT for IBS treatment needs to be clarified due to its advantage in terms of implementation in routine clinical practice.

Subgroup analyses revealed that bowel preparation may improve the efficacy of non-capsule FMT. These results are in line with previous findings that suggest bowel preparation can alter the fecal microbiota in healthy individuals [41,42] and with the last European consensus on FMT in clinical practice, that recommends bowel preparation before FMT [34]. 

Finally, we found that IBS subtype significantly influences the FMT efficiency delivered through colonoscopy, gastroscopy and nasojejunal tube. This may be related to the different characteristics between IBS subtypes, both in terms of clinical manifestations and in terms of microbiota alteration. Thus, some consideration should be given to stratifying randomized controlled trials by IBS subtype, to clarify whether the existence of constipation symptomology influences the efficacy of FMT.

### 4.2. Global Considerations about Randomized Controlled Trials Evaluating the FMT Efficacy on IBS Treatment

In clinical trial design, one methodological consideration that may affect the efficacy of FMT in IBS treatment, and possible cause of the heterogeneity yielded, is the lack of standardization in the recruited patients. Only two of the RCTs included in our review [31,32] considered the diagnosis of small intestinal bacterial overgrowth (SIBO) in the exclusion criteria. SIBO causes GI symptoms such as abdominal pain, bloating, gas, distension, flatulence, and diarrhea, that can lead to an incorrectly diagnosed IBS [43]. Likewise, PI-IBS that frequently occurs after an episode of infectious gastroenteritis [44], and may have a different microbiota signature [45], was only considered as an exclusion criteria in two trials [32,34]. Aroniadis et al. [31] revealed a trend toward greater improvement in PI-IBS patients who received FMT, according to a post-hoc analysis.

Also important, only four studies [31,32,33,34] excluded participants supplemented with probiotics prior to FMT and none gave specific instructions regarding their diet during the follow-up time, other than to keep it stable. Indeed, only one study [36] reported changes in dietary habits during the follow-up. Diet can affect many aspects of gut physiology such as motility, permeability, microbiome, visceral sensation, brain-gut interactions, immune regulation and neuro-endocrine function [46], thus being a relevant confounding variable. Participants’ background diet and change in diet during intervention should be reported to exclude any effect on IBS symptoms.

Regarding medication, only one study [32] excluded patients that were under concomitant IBS medication, such as antimotility, antispasmodic and antidepressant drugs [6]. As medication may mask IBS symptoms and affect gut microbiota composition [47], its intake should be monitored before FMT and during the follow-up. 

### 4.3. Strengths and Limitations of This Study

Several limitations in this systematic review should be acknowledged. First, our analyses are limited by the low number of available studies and the quality of the reported data. Second, most studies were performed in Europe, limiting generalizability. Third, in order to perform subgroup analysis, we simply divided FMT dosage and IBS subtypes into two groups, we did not have access to raw data from all studies. Fourth, we did not assess the impact on quality of life. This aspect was already addressed in a previous meta-analysis [18] that found a significant improvement in quality of life in IBS patients 12 weeks after FMT. Fifth, the study populations diversity may also have contributed to the heterogeneity of the results. For instance, some studies included patients with different disease severities and patients had a wide age range.

Despite all limitations, our systematic review brings a comprehensive overview of the methodological limitations in clinical trial design, as well as differences in FMT interventions performed to date. Thus, this review highlights FMT procedure variables that could contribute to the contradictory effects on FMT efficacy.

### 4.4. Future RCTs Assessing FMT Efficacy for IBS Treatment

Future RCTs may benefit from FMT donations prepared in stool banking centers under GMP conditions, but also from a stratification by IBS subtype and disease severity. Thus, the RCTs should exclude participants with the diagnosis of confounding diseases, as well as considering background diet and change in diet during intervention and the use of medication before and during the follow-up period. More well-designed RCTs are needed to firstly assess whether the efficacy of capsules made of materials that allow the delivery of FMT to a specific intestinal location, is comparable with other delivery methods. Secondly, the impact of stool formulation (fresh, liquid frozen or even lyophilized), the FMT dose-response and lastly, the bowel lavage preparation on FMT efficacy must be investigated. This knowledge will allow optimization of the FMT procedure and thus assess its true clinical potential for IBS patients and beyond.

## 5. Conclusions

The present systematic review with meta-analysis shows that IBS patients may benefit from FMT when administered via gastroscopy or nasojejunal tube and that FMT is overall safe for IBS. Furthermore, IBS subtype and bowel lavage may play an important role in the response to FMT. 

## Figures and Tables

**Figure 1 jcm-12-01725-f001:**
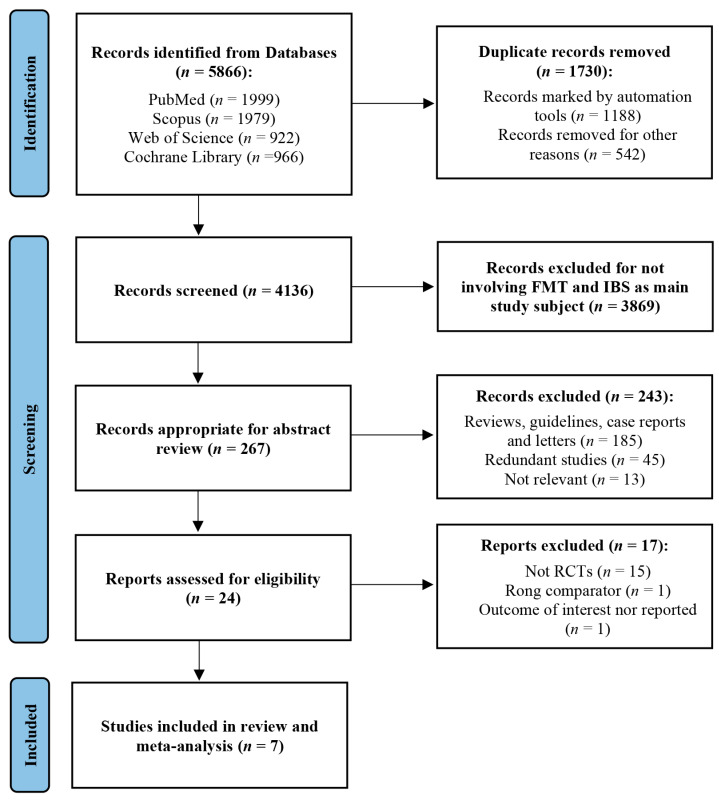
PRISMA study flow diagram describing the process of study selection.

**Figure 2 jcm-12-01725-f002:**
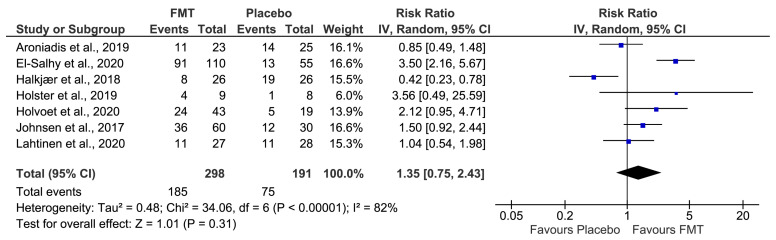
Forest plot of all studies for efficacy of FMT vs. placebo on global improvement of IBS symptoms. Aroniadis et al., 2019 [31], El-Salhy et al., 2020 [32], Halkjær et al., 2018 [33], Holster et al., 2019 [34], Holvoet et al., 2020 [35], Johnsen et al., 2017 [36] and Lahtinen et al., 2020 [37].

**Figure 3 jcm-12-01725-f003:**
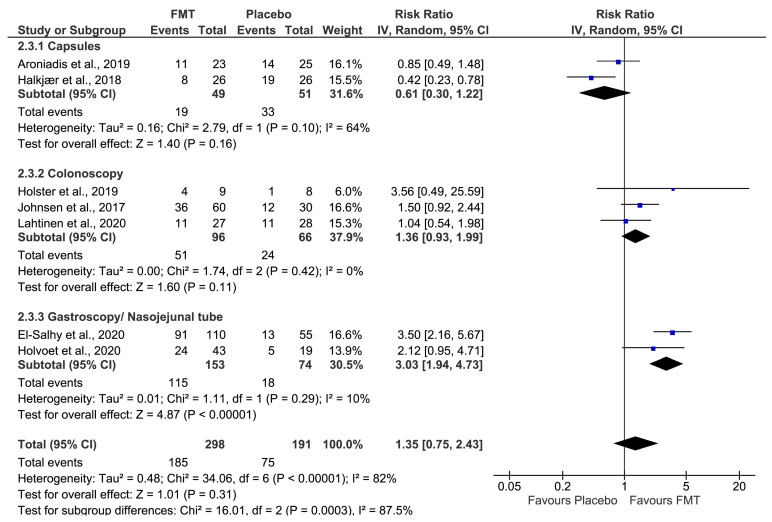
Forest plot for efficacy of FMT vs placebo on global improvement of IBS symptoms by delivery method. Aroniadis et al., 2019 [31], El-Salhy et al., 2020 [32], Halkjær et al., 2018 [33], Holster et al., 2019 [34], Holvoet et al., 2020 [35], Johnsen et al., 2017 [36] and Lahtinen et al., 2020 [37].

**Figure 4 jcm-12-01725-f004:**
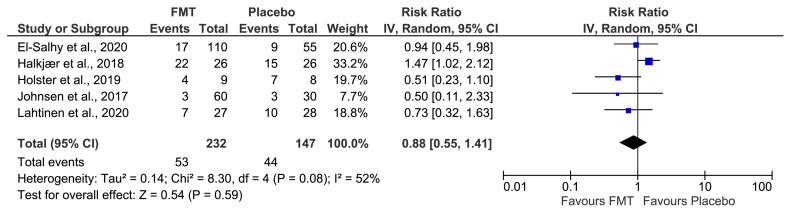
Forest plot of adverse events. El-Salhy et al., 2020 [32], Halkjær et al., 2018 [33], Holster et al., 2019 [34], Johnsen et al., 2017 [36] and Lahtinen et al., 2020 [37].

**Table 1 jcm-12-01725-t001:** Characteristics of the studies included in the systematic review.

Study	Country/Setting	Sample Size (*n* Analyzed)	Median or Mean Age ^1^/Years ^2^	% Females ^2^	Diagnostic Criteria Used for IBS and Subtypes ^2^ of IBS Recruited	FMT Route	Intervention/Dose	Control
Aroniadis et al., 2019 [31]	USA/Primary, secondary, and tertiary care (three centers)	48 (45)	I: 33 (27–48) C: 48 (28–48)	I: 36 C: 39	ROME III; 100% IBS-D	Oral capsules	75 FMT capsules containing 50 g feces from 1 of 4 donors	Placebo capsules not containing fecal microbiota
Halkjaer et al., 2018 [33]	Denmark/Tertiary care (two centers)	52 (46)	I: 37.3 (12.5) C: 35.5 (10.6)	I: 68 C: 69	ROME III; 33.3% IBS-C, 29.4% IBS-D, 37.3% IBS-M	Oral capsules	300 FMT capsules containing 144 g fecal matter derived from 600 g pooled donor feces (4 donors)	Placebo capsules not containing fecal microbiota
Holster et al., 2019 [34]	Sweden/Tertiary care (single center)	17 (16)	I: 34 (27–49) C: 39 (33–43)	I: 52 C: 65	ROME III; 25.0% IBS-C, 56.2% IBS-D, 18.8% IBS-M	Colonoscopy	30 g donor feces from 1 of 2 donors mixed with isotonic saline and 10% glycerol to a final volume of 150 mL	Autologous
Johnsen et al., 2017 [36]	Norway/Primary care (single center)	90 (83)	I: 44 (33–54) C: 45 (34–57)	I: 65 C: 68	ROME III; 53.0% IBS-D, 47.0% IBS-M	Colonoscopy	50–80 g pooled donor feces (2 donors) mixed with 200 mL isotonic saline and 50 mL 85% glycerol	Autologous
Lahtinen et al., 2020 [37]	Finland/Tertiary care (three centers)	55 (49)	I: 47.3 (16.8) C: 46.3 (14.3)	I: 52 C: 65	ROME IV; 51% IBS-D, 4.3% IBS-M, 34.7% IBS-O or IBS-U	Colonoscopy	30 g donor feces (1 donor) homogenized in 100–200 mL of water	Autologous
El-Salhy et al., 2020 [32]	Norway/Tertiary care (single center)	165 (164)	I_60_: 39.3 (13.2) I_30_: 39.2 (12.4) C: 41.2 (13.7)	I: 79 C: 85	ROME IV; 37.8% IBS-C, 38.4% IBS-D, 23.8% IBS-M	Gastroscopy	30 g and 60 g donor feces (1 single “super donor”) mixed with 40 mL isotonic saline	Autologous
Holvoet et al., 2021 [35]	Belgium/Tertiary care (single center)	64 (62)	I: 40 (25–59) C: 36 (18–63)	I: 69 C: 41	ROME III; 100% IBS-D or IBS-M	Nasojejunal tube	50–80 g pooled donor feces (2 donors) mixed with isotonic saline and glycerol	Autologous

Abbreviations: C: control group; FMT: fecal microbiota transplantation; IBS: irritable bowel syndrome; IBS-C: constipation predominant IBS; IBS-D: diarrhea predominant IBS; IBS-M: IBS with mixed stool pattern; IBS-O: other, IBS in remission (not meeting the Rome III criteria at the baseline); IBS-U: unsubtyped IBS; I: intervention group; I_30_: intervention group with 30 g doses; I_60_: intervention group with 60 g doses; USA: United States of America; y: years. ^1^ Age are median (IQR), median (range), or mean (SD); ^2^ at baseline.

**Table 2 jcm-12-01725-t002:** Characteristics of the studies included in the systematic review.

Study	Frequency/Duration	Follow-Up	Primary Outcome	Secondary Outcomes	Main Findings	Risk of Bias
Aroniadis et al., 2019 [31]	25 capsules daily × 3 days ^1^	6 mo.	Decrease in IBS-SSS ≥ 50 points at 12 wk.	IBS-QOL, HADS, Bristol stool scale scores and microbiota profiles.	No significant differences in IBS symptoms improvement, QOL, depression, anxiety, stool consistency and microbiome profiles between intervention and control groups. Significant similarity ^2^ between the patient and donor microbiota 1 wk after FMT.	Unclear
Halkjaer et al., 2018 [33]	25 capsules daily × 12 days	6 mo.	Decrease in IBS-SSS ≥ 50 points at 12 wk.	IBS-QOL and microbiota diversity.	Significant improvement in IBS symptoms and QOL in the placebo group compared to the intervention group. Significant similarity^4^ between the patient and donor microbiota after FMT.	Unclear
Holster et al., 2019 [34]	Once	6 mo.	Decrease in gastrointestinal symptom rating scale-IBS of ≥30%	IBS-SSS, IBS-QOL, HADS, visceral sensitivity and microbiota composition.	No significant differences in IBS symptoms improvement, QOL, anxiety and visceral sensitivity between intervention and control groups. No significant similarity ^2^ between the patient and donor microbiota after FMT.	Low
Johnsen et al., 2017 [36]	Once	12 mo.	Decrease in IBS-SSS > 75 points at 12 wk.	Decrease in IBS-SSS > 75 points at 12 mo.	Significant improvement in IBS symptoms in the intervention group compared to the control group.	Low
Lahtinen et al., 2020 [37]	Once	12 mo.	Decrease in IBS-SSS ≥ 50 points at 12 wk.	IBS-QOL, BDI, BAI, microbiota composition and fecal water content.	No significant differences in IBS symptoms improvement, QOL, depression, anxiety and stool consistency between intervention and control groups. Significant similarity ^4^ between patient and donor microbiota after FMT at all points after intervention, significantly higher in the intervention group compared to the control group.	Unclear
El-Salhy et al., 2019 [32]	Once	4 mo.	Decrease in IBS-SSS ≥ 50 points at 12 wk.	IBS-QOL, FAS, SF-NDI, dysbiosis index and microbiota profiles.	Significant improvement in IBS symptoms, QOL, fatigue and dyspepsia in the intervention group compared to the control group. Significant changes in microbiota abundance ^3^ in the intervention group but not in the placebo group.	Low
Holvoet et al., 2021 [35]	Once ^1^	12 mo.	Improvement in overall symptoms and abdominal bloating at 12 wk.	IBS symptom scores by using daily diary, IBS-QOL and microbiota composition.	Significant improvement in IBS symptoms and QOL in the intervention group compared to the control group. No significant similarity between the patient and donor microbiota after FMT.	High

Abbreviations: BDI: Beck Depression Inventory; BAI: Beck Anxiety Inventory; FAS: Fatigue Assessment Scale; FMT: fecal microbiota transplant; HADS: Hospital Anxiety and Depression Scale; IBS-QOL: irritable bowel syndrome—Quality of Life Questionnaire; IBS-SSS: irritable bowel syndrome—Severity Symptom Scale; mo.: months; QOL: quality of life; SF-NDI: Short-Form Nepean Dyspepsia Index; VAS: Visual Analogue Score; wk.: weeks; mo.: months; ^1^ First intervention of a crossover study; ^2^ Jensen-Shannon Distance; ^3^ GA-map Dysbiosis; ^4^ Mann-Whitney U test.

**Table 3 jcm-12-01725-t003:** GRADE summary of evidence on the efficacy of FMT in IBS by administration method.

Components	Nº of Participants	FMT	Placebo	Relative Effect (95% CI)	Absolute Effect (95% CI)	Certainty of the Evidence	Importance
Overall symptoms improvement	489 (7 RCTs)	185/298 (62.1%)	75/191 (39.3%)	RR 1.35 (0.75 to 2.43)	137 more per 1000 (from 98 fewer to 562 more)	⊕◯◯◯ Very low ^a,b,c^	CRITICAL
Symptoms improvement via oral capsules	100 (2 RCTs)	19/49 (38.8%)	33/51 (64.7%)	RR 0.61 (0.30 to 1.22)	252 fewer per 1000 (from 453 fewer to 142 more)	⊕◯◯◯ Very low ^a,b,c^	CRITICAL
Symptoms improvement via colonoscopy	162 (3 RCTs)	51/96 (53.1%)	24/66 (36.4%)	RR 1.36 (0.93 to 1.99)	131 more per 1000 (from 25 fewer to 360 more)	⊕⊕⊕◯ Moderate ^a^	CRITICAL
Symptoms improvement via gastroscopy or nasojejunal tube	227 (2 RCTs)	115/153 (75.2%)	18/74 (24.3%)	RR 3.03 (1.94 to 4.73)	494 more per 1000 (from 229 more to 907 more)	⊕⊕◯◯ Low ^a,b^	CRITICAL

Abbreviations: CI: confidence interval; FMT: fecal microbiota transplantation, IBS: irritable bowel syndrome, RCT: randomized controlled trial; RR: risk ratio. GRADE Working Group grades of evidence. High certainty: we are very confident that the true effect lies close to that of the estimate of effect. Low certainty: our confidence in the effect estimate is limited: the true effect may be substantially different from the estimate of the effect. Very low certainty: we have very little confidence in the effect estimate: the true effect is likely to be substantially different from the estimate of the effect. ^a^ Downgraded one level due to imprecision. ^b^ Downgraded one level due to risk of bias. ^c^ Downgraded one level due to inconsistency.

**Table 4 jcm-12-01725-t004:** Subgroup analyses of comparisons of FMT vs placebo in IBS.

	No. of RCTs	No. of Patients	RR (95% CI)	I^2^	*p* ^1^
All studies	7	489	1.35 (0.75–2.43)	82%	
Method of administration					0.0003
Capsules [31,33]	2	100	0.61 (0.30–1.22)	64%	
Colonoscopy [34,36,37]	3	162	1.36 (0.93–1.99)	0%	
Gastroscopy/Nasojejunal tube [32,35]	2	227	3.03 (1.94–4.73) **	10%	
Total dose					0.76
≥50 g [32,33,36]	3	252	1.35 (0.42–4.37)	93%	
<50 g [31,32,34,35,37]	5	292	1.67 (0.89–3.15)	74%	
FMT sample preparation					0.44
Fresh [34,35]	2	79	2.28 (1.09–4.78) *	0%	
Frozen [31,32,33,37]	4	320	1.08 (0.43–2.72)	90%	
Both [36]	1	90	1.50 (0.92–2.44)	-	
Bowel preparation					0.80
With bowel preparation [33,34,35,36]	4	221	1.26 (0.53–2.97)	80%	
Without bowel preparation [31,32,36]	3	268	1.48 (0.58–3.75)	88%	
IBS subtypes					0.77
With constipation type [32,33,34]	3	234	1.61 (0.30–8.69)	93%	
Without constipation type [31,35,36,37]	4	255	1.25 (0.87–1.79)	32%	

Abbreviations: CI: confidence interval; FMT: fecal microbiota transplantation; GI: gastrointestinal; IBS: irritable bowel syndrome; No: number; RCTs: randomized controlled trials; RR: risk ratio. ^1^ Test for subgroup differences. * *p* < 0.05; ** *p <* 0.00001.

**Table 5 jcm-12-01725-t005:** Subgroup analyses of comparisons of FMT vs placebo in IBS between studies that delivered FMT through colonoscopy, gastroscopy and nasojejunal tube.

	No. of RCTs	No. of Patients	RR (95% CI)	I^2^	*p* ^1^
All studies	5	389	1.94 (1.17–3.22)	63%	
Total dose					0.8
≥50 g [32,36]	2	200	2.38 (0.96–5.87)	86%	
<50 g [32,34,35,37]	4	244	2.06 (1.10–3.88) *	61%	
FMT sample preparation					0.64
Fresh [34,35]	2	79	2.28 (1.09–4.78) *	0%	
Frozen [31,32,33,37]	2	220	1.94 (0.59–6.40)	89%	
Both [36]	1	90	1.50 (0.92–2.44)	-	
Bowel preparation					0.84
With bowel preparation [34,35,36]	3	169	1.70 (1.13–2.55) *	0%	
Without bowel preparation [32,37]	2	220	1.94 (0.59–6.40)	89%	
IBS subtypes					0.003
With constipation type [32,34]	2	182	3.50 (2.19—5.60) **	0%	
Without constipation type [35,36,37]	3	207	1.44 (1.02–2.04) *	0%	

Abbreviations: CI: confidence interval; FMT: fecal microbiota transplantation; GI: gastrointestinal; IBS: irritable bowel syndrome; No: number; RCTs: randomized controlled trials; RR: risk ratio. ^1^ Test for subgroup differences. * *p* < 0.05; ** *p* < 0.001.

## Data Availability

Not applicable. This study uses data already published and available in the cited references.

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
