# Peer review of "Procedures in Fecal Microbiota Transplantation for Treating Irritable Bowel Syndrome: Systematic Review and Meta-Analysis"

_jcm, 2023, doi:10.3390/jcm12051725_

Round 1

Reviewer 1 Report

1) General comments

Dr. Tânia Rodrigues and Dr. André Moreira-Rosário, et al. reviewed “Procedures in fecal microbiota transplantation for treating irritable bowel syndrome: systematic review and meta-analysis”. This article is informative and well presented. The reviewer has some comments.

The authors report subgroup analyses of the effect of FMT, but the number of RCTs appears to be small. Please carefully describe the results and Discussion.

1.     The reviewer recommends that the authors should revise the "Our meta-analysis revealed for the first time a set of critical steps that likely affect the efficacy of FMT as clinical procedure to treat IBS." in the Abstract. Please rewrite the abstract's conclusion to include future considerations without overemphasizing the results of the subgroup analyses in this Systematic review.

2.     Please move the future considerations listed in Conclusion in "page 19, line 417 to page 20, line 427" to Discussion as “4.4”. These sentences are very useful for future studies and are key points of this paper. Future RCTs in the Conclusions of the text of this article should be described shortened.

Author Response

Dear Reviewer,

Thank you very much for your comments and suggestions.

Regarding the subgroup analysis, the low number of RCTs is a limitation as highlighted by the Reviewer. In the 4.3 subchapter named by Strengths and Limitations of This Study, the low number of available studies and the quality of the reported data is appointed as the first limitation.

Nevertheless, heterogeneity of the RCTs makes subgroup analysis a useful tool, enabling to explore the heterogeneity in terms of tendencies based on prespecified subgroups of clinical importance.

Answer to commentary 1.

Considering the Reviewer’ commentary, we changed the Abstract’s conclusion to “Our meta-analysis revealed a set of critical steps that could affect the efficacy of FMT as a clinical procedure to treat IBS, nevertheless more RCTS are needed.” This alteration was introduced without affecting Abstract maximum word counts.

Answer to commentary 2.

Regarding Reviewer’ suggestion, we moved the Future studies consideration from Conclusion to a new subchapter in the Discussion, named by “4.4. Future RCTs assessing FMT efficacy for IBS treatment”.

Finally, the manuscript was reviewed by a native English-speaking colleague with  background in science field. His name was included in the Acknowledge section.

Reviewer 2 Report

Good work

This is an upcoming field in IBS management.

Unfortunately this meta analysis is limited by the studies it is able to look at after removing all the ones that did not meet criteria.

In addition to "n" being so low, multiple limitations as very nicely pointed out by authors , including improper exclusion of associated disease like SIBO, potentially not classifying IBS subtype correctly etc limit the utility of this research to be used on a larger scale.

Author Response

Dear Reviewer,

Thank you very much for your comments.

Indeed, the low number of RCTs is a limitation as highlighted by the Reviewer. Nevertheless, heterogeneity of the RCTs makes subgroup analysis a useful tool, enabling to explore the heterogeneity in terms of tendencies based on prespecified subgroups of clinical importance.

Despite that, we changed the Abstract’s conclusion to avoid overemphasizing the results.

Round 2

Reviewer 1 Report

1) General comments

Dr. Tânia Rodrigues and Dr. André Moreira-Rosário, et al. revised “Procedures in fecal microbiota transplantation for treating irritable bowel syndrome: systematic review and meta-analysis”.

Thank you for your reply. I read your responses for my questions.

Your answers are precisely good, and I understood your points of view in your study.

I appraise your investigation.